# Loneliness and the persistence of fear: Perceived social isolation reduces evaluative fear extinction

**Erica Hornstein, Lee Lazar, Naomi Eisenberger** *

Department of Psychology, University of California, Los Angeles, Los Angeles, California, United States of America

* neisenbe@ucla.edu

## Abstract

Loneliness has been linked to a host of harmful physical and mental health outcomes, detrimental effects that may stem from increases in threat-responding caused by altered fear learning in lonely individuals. In particular, the heightened threat-vigilance that is a hallmark of loneliness may augment the processes by which fear learning occurs, ultimately resulting in a greater number of perceived threatening cues in the environment. However, almost no research has examined how loneliness alters fear learning processes in humans. Here, we investigated the effect of loneliness on fear learning during an evaluative learning procedure in which participants (n = 782) were taught to associate fearful, positive, or neutral control stimuli with neutral images. Results showed that reduced extinction of evaluative fear associations occurred in high (vs. low) lonely individuals, but there was no difference in extinction of evaluative appetitive (also known as positive or reward) associations, suggesting this effect is specific to fear learning. In addition to shedding light on the link between loneliness and poor health, these results represent an important step forward in the growing understanding of the powerful impact of social bonds on fear learning processes.

## Introduction

Research has long-demonstrated the harmful effects of being disconnected from others. In addition to the distress it causes, loneliness—the perception of being socially isolated [1] has been linked to poor physical and mental health outcomes [2–4], diminished cognition [5–7], poor emotion regulation [8–10], and a host of other negative consequences. These detrimental effects may stem from increased wear-and-tear on the body due to the enhanced vigilance for threat, and accompanying increased physiological stress-responding, that is a hallmark of loneliness [11]. Indeed, while this loneliness-induced hypervigilance may serve to promote survival, preparing socially isolated individuals for the enhanced probability of threat that stems from being without the protection and care of others, it may come at a cost, potentially augmenting the process by which people learn about threats and increasing the number and strength of their fears. Although animal research has demonstrated that objective social

**Data Availability Statement:** The data collected for this work are accessible on the Harvard Dataverse under the title "Loneliness and persistence of fear: Perceived social isolation reduces evaluative fear extinction" (https://doi.org/10.7910/DVN/EYCQK7).

**Funding:** National Science Foundation RAPID
Grant award to to N.I.E. and E.A.H (NSF 2034809:
https://www.nsf.gov). The sponsors played no role
in study design, data collection/analysis, decision
to publish, or preparation of the manuscript.

**Competing interests:** The authors have declared
that no competing interests exist.

isolation, the closest animal correlate to loneliness, augments fear learning [12, 13], little research has investigated the relationship of perceived social isolation in humans. In the current work, we conducted a large-scale study to examine the impact of loneliness on the ways fears are learned in humans.

Evidence in support of the link between loneliness and fear learning can be found in animal work examining the effects of social isolation on threat sensitivity. Although objective social isolation in animals is not equivalent to the subjective social isolation experienced by lonely individuals, it provides a useful approach to understanding the effects of a lack of social ties on behavior and physiology [14, 15]. Specific to fear learning and responding, this body of research has demonstrated that compared to their non-isolated counterparts, socially isolated animals exhibit increased anxious behaviors, augmented fear acquisition, delayed fear extinction (the process by which the association between a certain cue and an expected aversive outcome is reduced:[16]), and persistent fear responses that continue even when no longer in the presence of a threat [12, 17, 18]. Moreover, animals who undergo social isolation either before [13, 19, 20] or after [21] experiencing a trauma have been demonstrated to be more likely to develop the dysfunctional and persistent fears that are characteristic of PTSD, indicating that social isolation may alter fear learning in such ways that contribute to extreme fears and fear disorders. Altogether, these findings indicate that socially isolated animals experience amplified fear responses. Based on these findings, it is important to examine whether loneliness in humans can also enhance fear responding.

The first study to investigate the effect of loneliness on fear learning revealed effects similar to those demonstrated in animals. In this pilot work, the results across 3 separate samples were remarkably consistent—individuals high in loneliness exhibited reduced fear extinction, relative to those low in loneliness [22]. Although these studies had relatively small sample sizes, these preliminary studies suggest that loneliness may augment fear by reducing its extinction, either through the development of fears that are more robust, and therefore resist extinction, or by impairing extinction learning itself. Interestingly, in all three studies, acute interventions designed to increase feelings of social connection (e.g., a reminder of a close other) mitigated these effects [22], suggesting that the perceived lack of social bonds was critical for attenuating extinction in lonely individuals, and that when lonely individuals were reminded of their social bonds, fear extinction was enhanced. However, this initial investigation was only preliminary, and the small sample sizes and complex extinction procedures (including the presence of reminders of close others, etc.) raise questions regarding how much this work can be interpreted to represent the effects of loneliness on fear learning on a larger scale. Nonetheless, these findings suggest the need for deeper investigation of the relationship between loneliness and fear learning.

Importantly, while little is known about the effects of loneliness on fear learning, much has been uncovered regarding the relationship between the opposite experience of feeling socially connected and fear learning. In particular, research on the role of social support during Pavlovian fear conditioning has revealed that the presence of social support reminders (in the form of images of close others or even just thoughts of close others) not only inhibit fear in the short-term, while they are present [23, 24] but also inhibit fear in the long-term, after they are removed [23, 25, 26] by enhancing fear extinction [27, 28]. This combination of effects is unique, as fear inhibitors typically only reduce fear in the short-term, but lead to no change or even augmented fear in the long-term based on their safety-signaling role within the Pavlovian framework [16, 29]. It is thought that social support reminders (and physical warmth, which was recently demonstrated to have the same pattern of effects: [30]) may confer this unique combination of short- and long-term fear inhibition due to the importance of social bonds for survival [31]. This survival-relevance enables social support reminders to signal access to

resources, care, and security that may reduce the aversive value of expected outcomes, as well as to engage neurobiological systems in the process of reinforcing and maintaining social bonds that overlap with the systems that underlie and drive fear learning (endogenous opioid system:[32]). In combination with adjacent findings that the experience of social exclusion—the acute experience of being rejected by others—leads to augmented conditional fear responding during fear acquisition that occurs not only for the stimulus learned to be feared, but for similar cues (a process known as generalization: [33]), these findings add a piece to the puzzle regarding the impact of loneliness on fear learning. If feelings of *social connection* are able to have such powerful effects on reducing fear learning, it stands to reason that the contrasting, and equally survival-relevant, feelings of *social disconnection or loneliness* would have similarly powerful, fear-enhancing effects.

Together, the growing understanding of the powerful inhibitory role of experiences of social connection during fear learning combined with the above-mentioned preliminary evidence that experiences of loneliness have contrasting effects (reducing fear extinction: [22]) point to a likely augmenting effect of loneliness on fear learning via either more robust fear acquisition and/or impaired fear extinction. Examination of these ideas is important not only for continuing to build understanding of the effects of social bonds on fear learning, but also for elucidating whether these effects contribute to the higher risk for poor mental and physical health outcomes in lonely individuals by increasing threat-vigilance. Thus, it is critical to examine whether the extinction-reducing effects of loneliness that have been demonstrated in early preliminary work not only replicate but also occur in larger-scale investigations and in settings outside of the laboratory.

Therefore, in the current work we sought to examine the impact of loneliness on evaluative fear extinction in a large sample. Evaluative learning is similar to traditional fear learning in both underlying associative processes as well as learning procedures; however, it measures changes in affective value or liking instead of changes in automatic fear responding (which often involve physiological measurement). Specifically, evaluative learning engages the same associative processes as traditional fear learning (Pavlovian conditioning), during which a change in response to a target cue occurs after the target cue is repeatedly paired with a specific outcome [34, 35]; for example, the response to viewing an image might change from a neutral response to a fearful one (e.g., less liking, higher arousal) after the image is paired with a painful outcome (e.g., shock). However, while traditional fear learning assesses this change in responding via a multitude of fear-response outcomes, such as increased physiological arousal (skin conductance) or avoidance responses such as eye-blinking [36], evaluative fear learning assesses this change in responding solely via evaluative—also described as affective—responding [35]. In particular, evaluative learning assesses changes in liking or levels of positive and negative affect in response to a target cue. In this fashion, evaluative fear extinction outcomes are assessed by looking at reductions in acquired evaluative responses (e.g., liking, positive/negative affect) after a target cue has been repeatedly presented in the absence of a fearful outcome (extinction procedure) [35]. Although this form of conditioning relies on reported feelings, not the physiological responses that are typically measured in in-person Pavlovian conditioning procedures, the learning that occurs during these evaluative procedures has been shown to occur reliably [35] and to run parallel to those that occur during Pavlovian procedures [37, 38]. Importantly, recent work using computational modeling has demonstrated that changes in learned expectancy ratings, which typically track with physiological and neurological fear responding, occur at the same rate and learning curve during both online evaluative fear learning and in-person Pavlovian fear learning procedures [39]. Thus, here, we chose to implement online evaluative (rather than Pavlovian) extinction in order to assess a larger number of people outside of the lab setting.

Using a sample of individuals recruited from across the United States, we examined whether being high or low lonely led to differences in evaluative fear extinction (compared to evaluative appetitive (reward) extinction or simple repetition (no evaluative learning)). These procedures enabled us to directly examine whether there were differences in extinction of evaluative responses for target images previously paired with positive images or fearful images across low and high lonely individuals.

## Methods

### Participants

1005 (gender: 440 *female*, 565 *male*, 13 *other;* mean$_{AGE}$ 32 years old; ethnicity (per Prolific participant information): 700 *White*, 100 *Asian*, 75 *Black*, 130 *Mixed*).

**Recruitment.** Participants were recruited from across the United States through the online participant recruitment platform, Prolific (www.prolific.com: recruitment period from July 2$^{nd}$, 2020 to December 10$^{th}$, 2020) and compensated with cash payments through this platform. Participants were allowed to sign up for the study if they met the study criteria: they had to be between the ages of 18 and 55 years old and fluent in English.

**Consent.** All procedures were carried out on participants' home computers using Qualtrics surveys (https://qualtrics.com: links provided through the Prolific platform). Before beginning any study procedures, participants were first asked to read through and indicate agreement with an approved informed consent document (agreement indicated by clicking to enter the survey and non-agreement indicated by clicking to end the experimental procedures).

**Ethics statement.** All study procedures were approved by the University of California, Los Angeles Institutional Review Board (IRB #20–001098) and were carried out in accordance with relevant guidelines and regulations. Informed consent was received by asking participants to read through an informed consent document and to indicate their consent by confirming their prolific identification number and clicking to continue the survey.

### Procedures

This study was conducted as part of a larger study [see: 40], and therefore these procedures were conducted in the context of other protocols that were designed to examine the effects of a prosocial behavior intervention on feelings of well-being. However, for the procedures used to investigate the questions of interest for the current work, all participants underwent the same procedures.

After indicating consent, participants were asked to ensure that they completed experimental procedures in a quiet space during which they could focus on the questions and images being presented on the screen. Participants signed agreements to this effect before completing any tasks and reported on any distractions or issues they encountered while completing the tasks. Reporting of such issues was minimal, although several participants were unable to complete the study due to computer malfunction or internet connectivity. Due to the format of Prolific, these participants were not formally enrolled in the sample and none of their data was reported to the experimental team.

**Session 1.** After signing up, participants were asked to complete several questionnaires. These included assessments of their daily social interaction and living situations as well as their self-reported levels of loneliness (using the UCLA Loneliness Scale v3: [41]). It should be noted that as this data was collected during the first year of the COVID pandemic, the information regarding participants daily social interactions and living situations was collected to determine participants' levels of objective social isolation. We found that less than 6% of our

total sample met the criteria for objective social isolation (social distancing, under stay-at-home orders, and living alone: n = 59), and therefore individuals who were socially isolated were not well-represented in this sample. Although it would be highly unlikely that social isolation could be investigated or would be likely to have an effect with such small numbers, we did examine whether social isolation led to any differences in fear learning and found no such effects ($p > .05$).

Participants then underwent an evaluative acquisition procedure during which they viewed presentations of 9 target neutral images, each of which was consistently followed by presentations of a secondary image that was either positive (3), neutral (3), or fearful (3) in nature. Target neutral images depicted everyday objects (i.e., pen, water bottle, pillow, shopping basket) on a white background and all secondary images were taken from the IAPS data set. In particular, positive images were chosen to depict young animals (e.g., kittens, puppies, bunnies, etc.) that would trigger approach and liking reactions. Neutral images (for both the target images and the secondary images) were chosen to depict everyday items (e.g., pens, bowls, lamps, etc.) that participants would easily recognize and be familiar with, but that should elicit no emotional reaction. Finally, fearful images were chosen to depict threatening animals (e.g., snakes, spiders, sharks, etc.) that would trigger avoidance and fear reactions. There were three pairings for each pairing condition (positive-paired, neutral-paired, and fearful-paired) and each pairing was presented 7 times in a pseudo-randomized order (structured so that no more than 3 pairings of a certain type were presented consecutively) (see Fig 1A). Image pairings were counterbalanced such that each target neutral image was included in each pairing condition across three different randomly assigned versions of the acquisition procedure. Each image presentation was 1s in duration and between pairings there was a 4s inter-trial-interval. This procedure was designed to allow participants to acquire associations between each target image and the secondary image that followed.

After the evaluative learning procedure, participants were then presented with each of the 9 target neutral images on its own and asked to provide ratings for how much they liked the image (using a scale from 1 to 7, where 1 was "do not like at all" and 7 was "like very much")

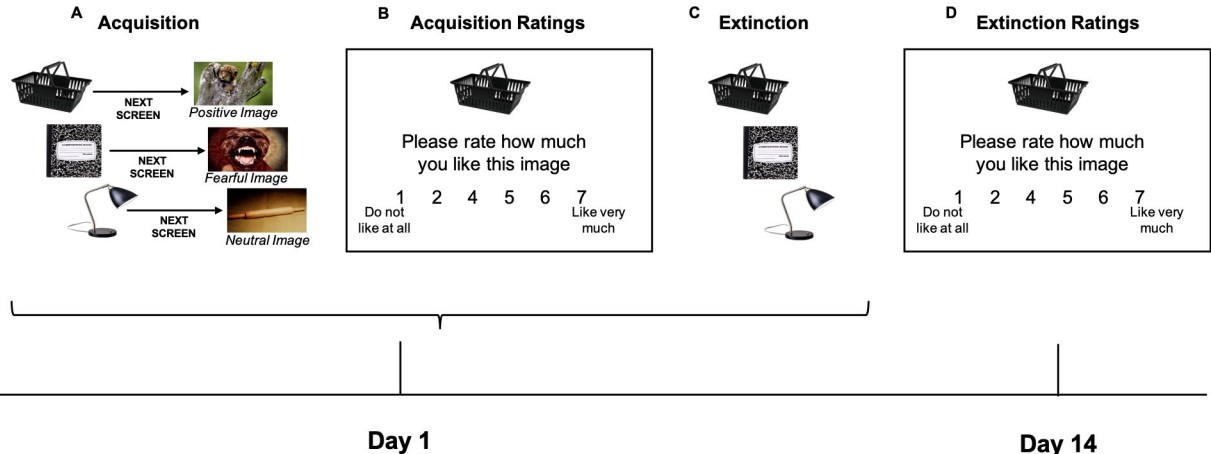

**Fig 1. Depiction of study procedures.** A. On day 1, participants completed an evaluative acquisition procedure during which target neutral images were presented on the screen, followed by a second screen on which was presented an image that was either positive, fearful, or neutral. B. Participants then rated how much they liked and how positive/negative they found each target neutral image following acquisition (only liking ratings depicted here), enabling assessment of whether the evaluative acquisition procedure was effective and whether evaluative associations formed. C. Participants then underwent an evaluative extinction procedure, during which they viewed each target neutral image with no secondary image following. D. On day 14, participant rated how much they liked and how positive/negative they found each image (only liking ratings depicted here), enabling assessment of changes in acquired evaluative associations following the extinction procedure.

and how positive or negatively they felt about the image (using a scale from 1 to 7, where 1 was "extremely negative" and 7 was "extremely positive") (see Fig 1B). These ratings enabled us to assess whether the target neutral images were associated with the following secondary images, indicated by the ability of the target images alone to produce feelings of liking or affective responses that reflect the nature of the associated secondary image.

Finally, participants underwent an evaluative extinction procedure during which they viewed presentations of each neutral target image, but this time no secondary image followed these presentations (see Fig 1C). There were 3 presentations of each target neutral image, each image presentation was 1s in duration followed by a 4s inter-trial-interval. This procedure was designed to allow participants to acquire new learning that the target neutral images are not always followed by the secondary images in order to weaken the associations acquired during the evaluative acquisition procedure (extinction).

Image presentations for the acquisition and extinction procedures were made using videos created with Microsoft PowerPoint that were then presented within the Qualtrics survey. Image presentations for the ratings were simply presented on Qualtrics and all ratings were made using Qualtrics response systems. Participants were unable to click forward on the Qualtrics screen until each video was complete and all ratings were required for participants to proceed through the experiment.

**Session 2.**   Two weeks after session 1, participants were contacted through Prolific with another link to a Qualtrics survey. Those who returned for this session (n = 782) were asked to complete the same ratings they had provided during session 1: rating how much they liked each target neutral image and how positive or negatively they felt about each target neutral image (see Fig 1D). These ratings enabled us to assess whether the previously learned associations between the target neutral images and the following secondary images were weakened during the evaluative extinction procedure, indicated by a return toward neutral for the liking and affective ratings in both the positive and fearful pairing conditions.

It should be noted that in the two weeks between sessions 1 & 2, participants were randomly assigned to one of three conditions: 1) completing 3 acts of kindness toward others each week (e.g., buying a cup of coffee for another, reaching out to check on a friend: prosocial behavior intervention), 2) completing three acts of kindness toward themselves each week (e.g., buy a cup of coffee for themselves, take time to read a favorite book: self-kindness intervention), or 3) completing no specific acts each week (control). Although these interventions were implemented for the larger study design, they still had the potential to influence expression of extinction learning during session 2 and therefore are being reported here. Although examination of the interaction of intervention and fear extinction outcomes (intervention condition assignment x procedure x condition) revealed no significant effects (p = .575), we added intervention condition assignment as a covariate to our investigations of the effects of loneliness in order to account for the fact that participants underwent different assigned conditions between timepoints 1 & 2.

## Analyses

Before analyzing the results, participants were split into low and high lonely categories based on loneliness cutoffs taken from previous work [22, 42]. In particular, participants who scored equal to or below 40 on the UCLA Loneliness Scale (total scale ranging from 20–80: [41]) were considered to be low lonely, while those who scored equal to or above 41 were considered to be high lonely.

In order to analyze evaluative learning, we first averaged across the ratings for each of the 3 target neutral images in each pairing condition to create a master score. Therefore, for each

participant we had one master score for the positive-paired condition, the neutral-paired condition, and the fearful-paired condition for both acquisition and extinction.

Next, in order to assess whether evaluative associations were acquired and the effect of loneliness on this acquisition, we examined ratings across the pairing conditions and low and high lonely groups. We conducted a 2 (loneliness: low or high) x 3 (pairing condition: positive-paired, neutral-paired, or fearful-paired) mixed ANOVA for both the liking ratings and the affect ratings as well as priori planned post-hoc tests (with Bonferroni corrections) to assess differences across pairing conditions. This allowed us to assess interactions of loneliness and pairing condition on evaluative acquisition as well as main effects of both loneliness and the pairing conditions.

Finally, in order to assess whether evaluative extinction occurred and any effects of loneliness on this extinction, we examined ratings across procedures (in order to investigate any change in these ratings from acquisition to extinction) across pairing conditions and the low and high lonely groups. We conducted a 2 (procedure: acquisition or extinction) x 2 (loneliness: low or high) x 3 (pairing condition: positive-paired, neutral-paired, or fearful-paired) mixed ANCOVA for both the liking ratings and the affect ratings, with intervention condition assignment for the larger study (prosocial intervention, self-kindness intervention, control) added as a covariate. The results reported above include this covariate; however, if we exclude the intervention condition assignment covariate from these analyses, the pattern of significant results does not change (please see S1 File for these analyses). Overall, these analytic procedures allowed us to assess interactions of loneliness and pairing condition on evaluative extinction.

## Results

### Evaluative fear acquisition

Results showed that while evaluative acquisition did occur, there was no difference in evaluative acquisition across low (n = 357) and high (n = 648; samples reflect all participants who completed Day 1 (results from only those who returned for Day 2 follow the same pattern of effects: see S1 File); In particular, a comparison of evaluative responding (liking ratings) for each condition (positive-paired, neutral-paired, fear-paired) across low vs. high lonely groups revealed no interaction ($F(1.701,1705.715) = .947$, $p = .376$, $\eta^2 = .001$: Greenhouse-Geisser reported as Mauchly's test had a $p < .01$), and only a main effect of pairing condition ($F(1.701,1705.715) = 94.287$, $p < .001$, $\eta^2 = .086$: Greenhouse-Geisser reported as Mauchly's test had a $p < .01$), such that positive-paired target images were rated more highly liked than either neutral-paired ($p < .001$, 95% CI [.177, .304]) or fearful-paired ($p < .001$, 95% CI [.350, .532]) target images, and neutral-paired target images were more highly rated than fear-paired target images ($p < .001$, 95% CI [.127, .274]: Bonferroni corrections applied on all pairwise comparisons) (please see Fig 2). Interestingly, results of the affect ratings followed the same patterns, but were not significant (please see the S1 File for full analysis of affect ratings). This is likely due to the more objective nature of affect ratings vs the more subjective nature of liking ratings; for example, rating an everyday, neutral object like blue cup as highly positive may feel odd, given that others may not be perceived as experiencing it as positive, while rating it highly liked may feel more appropriate, as liking is an individual experience. Therefore, because participants may have been taking into account their perceptions of others' views when it came to what is negative or positive, their reports on the affect scale may have been less influenced by the evaluative conditioning.

It is important to note that the effect size for these results, here and throughout the analyses below, is relatively low ($\eta^2 \leq .086$), which is likely due to the large amount of noise in this data

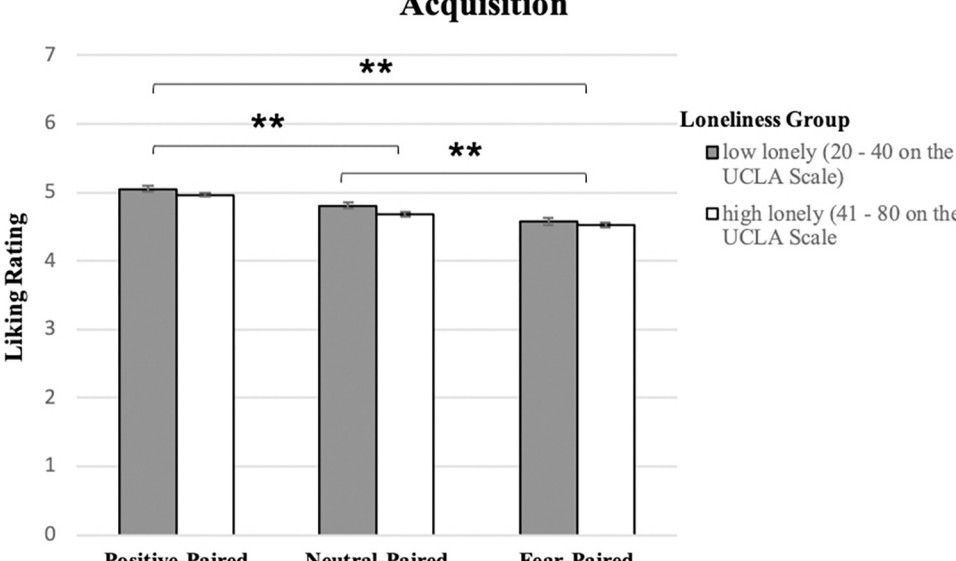

**Fig 2. Results comparing liking ratings (1, "do not like at all", to 7, "like very much") for images in each condition in both low and high lonely samples show differences in liking ratings across conditions, indicating evaluative learning occurred, but no difference across loneliness groups ** indicates p < .001.** All error bars reflect standard error.

due to participants completing these procedures in a wide variety of settings (outside of the lab), with an unknown array of distractions (e.g., noise in the home), possible technological differences (e.g., differing quality computer screens), and differing levels of attentional engagement. These differences in experimental setting and experience may have contributed for the low effect sizes being seen here. While this outcome certainly limits the interpretability of these results, we still believe the significant findings and patterns of effects demonstrated here are worthwhile contributions to the understanding of the impact of social bonds on fear learning.

Altogether, these results indicate that evaluative appetitive responses were acquired in the positive-paired condition, indicated by higher liking scores compared the neutral-paired condition, and evaluative fear responses were acquired in the fearful-paired condition, indicated by lower liking scores compared to the neutral-paired condition. Furthermore, evaluative responding was distinct across evaluative learning condition as would be expected, with higher liking for the positive-paired vs fearful-paired conditions, and there were no differences in this learning as a function of loneliness.

### Evaluative fear extinction

Once it was determined that evaluative acquisition had occurred, it was possible to examine the strength of extinction by examining whether evaluative responses were reduced from acquisition to extinction and whether this varied as a function of loneliness. A comparison of evaluative responding (liking ratings) for each condition across low (n = 308) and high (n = 474; these numbers reflect participants who completed both day 1 & day 14) lonely groups during acquisition (directly following acquisition procedures) and extinction (two-week follow-up post extinction procedures) revealed an overall significant interaction of loneliness (high, low) x procedure (acquisition, extinction) x condition (positive-paired, neutral-paired, fearful-paired) ($F(1.853,1443.692) = 3.411$, $p = .037$, $\eta^2 = .004$: Greenhouse-Geisser reported

## Extinction

**Fig 3. Results comparing liking ratings (1- do not like at all, 7 –like very much) for images in condition from directly post acquisition (timepoint 1) to two weeks post extinction (timepoint 2) in both low and high lonely samples.** ** indicates p < .001, * indicates p < .05, "ns" indicated p > .05. All error bars reflect standard error.

for Mauchly's test had a *p < .01*). Further investigation revealed that, as expected, appetitive extinction occurred in both low and high lonely individuals, indicated by decreased liking for positive-paired images from acquisition to extinction (low lonely: t(307) = 3.829, *p < .001*, 95% CI [.054, .10]; high lonely: t(473) = 6.089, *p < .001*, 95% CI [.18, .35]: Bonferroni corrections applied on all comparisons). In the neutral-paired condition, there was no change in responding from acquisition to extinction in the low lonely group (*p = .246*), reflecting the lack of acquired evaluative response to be reduced, but there was a decrease in liking in the high lonely group (t(473) = 2.575, *p = .01*, 95% CI [.04, .03]: Bonferroni correction applied) (this was likely due to the tendency of lonely individuals to experience events as increasingly less positive and pleasurable over time: [43]). Critically, in the fearful-paired condition, extinction of evaluative fear responses occurred only in the low lonely condition, indicated by increased liking from acquisition to extinction (t(307) = -4.006, *p < .001*, 95% CI [-.38, -.13]: Bonferroni correction applied)). However, no such extinction occurred for high lonely individuals in the fearful-paired condition (change from acquisition to extinction was not significant: *p = .689*), indicating that, as shown in our preliminary work, extinction of fear is reduced in high lonely individuals (see Fig 3).

Analyses further revealed a main effect of condition (F(1.867,1454.342) = 9.675, *p < .001*, $\eta^2$ = .012: Greenhouse-Geisser reported as Mauchly's test had a *p < .01*), such that positive-paired target images were rated more liked than either neutral-paired (*p < .001*, 95% CI [.129, .244]) or fear-paired target images (*p < .001*, 95% CI [.196, .338]), and neutral-paired target images were rated more liked than fearful-paired target images (*p = .004*, 95% CI [.020, .141]: Bonferroni corrections applied on all comparisons), indicating that, as found in the acquisition data, the learning procedure was successful in training participants to associate positive, neutral, or fearful associations with target images and similar patterns existed post extinction. These results also revealed a main effect of loneliness (F(1, 779) = 7.349, *p = .007*, $\eta^2$ = .009), such that high lonely individuals exhibited overall lower liking ratings compared to low lonely individuals (*p = .007*, 95% CI [.039, .245]: Bonferroni correction applied)—likely an effect of the more overall decreased positivity and pleasure reported by lonely individuals [43, 44].

Finally, analyses revealed an interaction of loneliness x procedure (acquisition vs extinction: F(1,779) = 3.976, *p = .046*, $\eta^2$ = .005), such that liking values exhibited a greater overall decrease from acquisition to extinction (regardless of pairing condition) in high lonely individuals (M$_{decrease}$ = .121) compared to relatively little change in low lonely individuals (M$_{decrease}$ = .006). This is likely another reflection of the tendency of lonely individuals to be disposed toward

negativity and displeasure that can create a cycle of increasing negativity and displeasure over time [43]. Analyses also revealed a significant interaction of condition x procedure (acquisition vs extinction: $F(1.853,1443.692$: Greenhouse-Geisser reported as Mauchly's test had a $p < .01$) $= 44.956, p < .001, \eta^2 = .054$), such that from acquisition to extinction, there were greater decreases in liking for the positive-paired target images ($M_{decrease} = .237$) compared to the neutral-paired target images ($M_{decrease} = .090$), indicating extinction of the evaluative appetitive associations as the positive associations are diminished and responses to the target images return toward a neutral value (appetitive extinction). There was also an increase in liking from acquisition to extinction in the fearful-paired condition compared to the neutral-paired condition ($M_{decrease} = -.138$), indicating extinction of the evaluative fear associations as the negative associations are diminished and the target images return toward neutral value (fear extinction). This interaction suggests that when high and low lonely groups are collapsed, typical patterns of positive extinction and fear extinction are observed.

## Discussion

Altogether, these results replicate those from preliminary studies examining the effects of loneliness on fear learning, all of which demonstrate a similar pattern of reduced extinction of fear in lonely individuals [22]. In particular, the findings of the current work show that while lonely individuals exhibit evaluative appetitive extinction, just as their non-lonely counterparts do, they do not exhibit evaluative fear extinction, in contrast to their non-lonely counterparts. This repeated pattern of effects is notable because it suggests that lonely individuals experience altered fear learning that may result in increased fears. Ultimately, this increase in fear may contribute to amplified perceptions of threat in the environment and consequent threat-related stress, resulting in the wear-and-tear that is thought to link loneliness to poor physical and mental health outcomes [45].

While reduced fear extinction in lonely individuals has now been repeatedly demonstrated, it is not clear why these effects are occurring. One possibility is that lonely individuals acquire initial fears that are more robust, leading them to resist extinction and persist beyond extinction procedures. Although no study so far has detected a difference in fear acquisition across low and high lonely individuals [current work,22], these studies were designed to examine fear extinction, not fear acquisition, and thus the acquisition procedures applied were designed to encourage quick acquisition (i.e., the use of 100% reinforcement) and may not have enabled detection of acquisition differences. Thus, it is possible other acquisition procedures may reveal greater acquired fear in lonely individuals. It is possible that the robust characteristic of acquired fears in lonely individuals is not found in their expression, but in their endurance, and therefore different measures that test how whether these fears persist over time are required. Another possibility is that lonely individuals undergo weaker extinction learning, less readily acquiring the new learning that a cue is not always associated with a fearful outcome that is required for extinction to occur [16, 46, 47]. As these studies are the first to examine the relationship between loneliness and fear learning, future work is required to tease apart these separate possibilities to better pinpoint the mechanism underlying poorer fear extinction outcomes in lonely individuals.

Looking to other work considering similar outcomes or experiences may shed some light on the relationship between loneliness and reduced fear extinction. For example, individuals with anxiety disorders show heightened fear acquisition and impaired fear extinction [48]. It is thought that this altered fear learning is caused in large part by dysfunctional safety detection and threat responding processes [48, 49], as well as dysfunctional activity in a neural region associated with safety detection and threat response inhibition (ventromedial prefrontal

cortex: [49–51]). However, anxious individuals do not only exhibit reduced fear extinction, as lonely individuals do, but also exhibit inappropriate learning, such that they respond to safety signals as threatening and threatening cues as safe, indicating impaired ability to discriminate between safe and threatening cues [49, 52]. However, this pattern of responses is not shared by lonely individuals, who have been shown to exhibit appropriate discriminatory learning, such that they accurately associate fear to cues paired with a threat (i.e., shock) and safety to those never paired with a threat [22] and to similarly acquire appropriate evaluative learning as in the current work, such that they were able to appropriately associate positive and negative value to cues based on pairings in the acquisition procedure. Therefore, while the dysfunctions in safety detection and threat responding that underlies reduced fear extinction in anxious individuals may play a role in reduced fear extinction in lonely individuals, it is not the whole story.

Another source of insight may be the effects of the opposite experience of loneliness—social support—on fear extinction. Reminders of social support bring about enhanced fear extinction [23, 27, 28]. In our emerging model of the effects of social bonds on fear learning, we have suggested that these effects may be due to engagement of the endogenous opioid system which is both central to reinforcing and maintaining social bonds [53] and the negative feedback system that underlies associative changes occurring during fear learning [16, 32]. It is notable that loneliness, which is associated with a lack of social support, has contrasting effects on the endogenous opioid system: whereas social support increases endogenous opioid activity [53], loneliness decreases endogenous opioid activity [54]. Correspondingly, separate work from the fear learning literature has revealed that increased opioid activity leads to reduced fear acquisition and enhanced fear extinction [32, 55, 56], while reduced opioid activity leads to augmented fear acquisition and reduced fear extinction [56–58]—a similar pattern of effects found for social support and loneliness, respectively. Thus, it may be that the ability of social connection and disconnection processes to influence endogenous opioid activity is what enables them to modulate the neurobiological systems underlying learning processes. Yet, while the effects of loneliness demonstrated in the current work fit nicely into this emerging model, this is only the first study of its kind and further examination of the possible involvement of opioids in this process is required.

Regardless of the pathway by which it occurs, the link between loneliness and reduced fear extinction is a crucial one of which to be aware. Loneliness has been linked with both symptom onset and progression in individuals who suffer from anxiety disorders [15, 59–63]. Notably, the most effective treatments to date for these disorders, exposure therapies, rely on fear extinction processes to reduce fear symptoms [49], and therefore may be less effective in those who are lonely. Compounded with the impaired extinction effects already found in anxious individuals, those who are anxious and lonely may be even more vulnerable to chronic fear symptoms and relapse. A such, methods to mitigate the effects of loneliness on fear extinction may be especially useful in overcoming extinction impediments and boosting exposure therapy outcomes. For example, in preliminary work, the presence of a social support reminders, which may increase feelings of social connection, was shown to enhance fear extinction outcomes in lonely individuals such that fear extinction occurred and followed the same pattern as fear extinction in non-lonely individuals [22]. However, these results were only shown in small samples and in non-anxious individuals. Therefore, before it is possible to determine if social support reminders can be leveraged to improve treatment outcomes in lonely individuals, more work is needed to examine if these findings replicate in a well-powered sample as well as in participants diagnosed with anxiety.

It is important to note that while the current study replicated previous patterns of effects [22], in order to examine these processes in a large group of individuals, it was necessary to use

online evaluative learning procedures. Therefore, further direct examination of whether these effects replicate in in-person fear learning procedures will enable the assessment of physiological responding in addition to subjective reports and to determine whether the effects of loneliness on fear learning persist. Additionally, although we found no effect of the self-kindness of pro-social interventions used for the larger study on learning outcomes, future work that does not include these interventions will remove any possibility that they influenced results.

Although in its infancy, research investigating the role of loneliness in fear learning holds the promise to have large impacts. Unfortunately, rates of loneliness have been on the rise [64, 65] with roughly 60% of Americans reporting feeling lonely in both 2020 [66] and 2022 [67], numbers that are reflected in countries across the world [68]. Therefore, progress in identifying the links between loneliness and well-being is critical. The results demonstrated in the current work represent an important step in deepening understanding of the link between loneliness and poor physical and mental health outcomes as well as elaborating the model of the effects of social bonds on fear learning and how these effects can be leveraged to improve treatment outcomes. Perhaps most importantly, the findings discussed in this paper have the potential to inform future investigations of potential interventions targeted at mitigating the harmful effects of loneliness. It is possible that in-the-moment reductions in loneliness during fear reduction procedures or threatening events may reduce both the unnecessary fears that may be maintained in lonely individuals, and the dysfunctional fears that contribute to the link between loneliness and fear disorders.

## Supporting information

**S1 File.**
(DOCX)

## Author Contributions

**Conceptualization:** Erica Hornstein, Naomi Eisenberger.

**Data curation:** Erica Hornstein, Lee Lazar.

**Formal analysis:** Erica Hornstein.

**Funding acquisition:** Erica Hornstein, Naomi Eisenberger.

**Investigation:** Erica Hornstein, Lee Lazar.

**Methodology:** Erica Hornstein, Lee Lazar, Naomi Eisenberger.

**Project administration:** Erica Hornstein, Lee Lazar.

**Resources:** Naomi Eisenberger.

**Supervision:** Erica Hornstein, Naomi Eisenberger.

**Validation:** Erica Hornstein.

**Visualization:** Erica Hornstein.

**Writing – original draft:** Erica Hornstein, Naomi Eisenberger.

**Writing – review & editing:** Erica Hornstein, Naomi Eisenberger.

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
