## [Decision Letter · Decision Letter 0]

19 Feb 2024

PONE-D-24-02974Loneliness and the persistence of fear: Perceived social isolation reduces evaluative fear extinctionPLOS ONE

Dear Dr. Eisenberger,

Thank you for submitting your manuscript to PLOS ONE. After careful consideration, we feel that it has merit but does not fully meet PLOS ONE’s publication criteria as it currently stands. Therefore, we invite you to submit a revised version of the manuscript that addresses the points raised during the review process.

The manuscript has elicited positive feedback from both reviewers, who have acknowledged the commendable execution of the study and the articulation of the paper. Nonetheless, both reviewers have highlighted significant concerns regarding the manuscript's positioning within the extensive body of literature on loneliness and fear conditioning. A particular point of contention is the methodological approach adopted in the study, notably the preference for subjective reports over objective measures such as skin conductance response. This methodological choice has been identified as a potential weakness by one reviewer, a critique that finds some justification within the traditional animal literature context. However, it is also pertinent to note that recent discussions within the field have begun to challenge the validity of fear conditioning as a reliable measure of fear. The reliance on subjective reports for assessing both loneliness and fear conditioning/extinction introduces complex issues that merit careful consideration. We strongly encourage the authors to address these critical points, alongside the detailed feedback provided by the reviewers, to enhance the manuscript's contribution to the field.

We look forward to receiving your revised manuscript.

Kind regards,

Rei Akaishi

Academic Editor

PLOS ONE

Journal Requirements:

"This work was funded by a National Science Foundation RAPID Grant (NSF 2034809) awarded to E.A.H. & N.I.E."

"National Science Foundation RAPID Grant award to to N.I.E. and E.A.H (NSF 2034809: https://www.nsf.gov). The sponsors played no role in study design, data collection/analysis, decision to publish, or preparation of the manuscript. "

4. In this instance it seems there may be acceptable restrictions in place that prevent the public sharing of your minimal data. However, in line with our goal of ensuring long-term data availability to all interested researchers, PLOS’ Data Policy states that authors cannot be the sole named individuals responsible for ensuring data access (http://journals.plos.org/plosone/s/data-availability#loc-acceptable-data-sharing-methods).

5. PLOS requires an ORCID iD for the corresponding author in Editorial Manager on papers submitted after December 6th, 2016. Please ensure that you have an ORCID iD and that it is validated in Editorial Manager. To do this, go to ‘Update my Information’ (in the upper left-hand corner of the main menu), and click on the Fetch/Validate link next to the ORCID field. This will take you to the ORCID site and allow you to create a new iD or authenticate a pre-existing iD in Editorial Manager. Please see the following video for instructions on linking an ORCID iD to your Editorial Manager account: https://www.youtube.com/watch?v=_xcclfuvtxQ.

Additional Editor Comments :

Editorial Feedback

The manuscript has elicited positive feedback from both reviewers, who have acknowledged the commendable execution of the study and the articulation of the paper. Nonetheless, both reviewers have highlighted significant concerns regarding the manuscript's positioning within the extensive body of literature on loneliness and fear conditioning. A particular point of contention is the methodological approach adopted in the study, notably the preference for subjective reports over objective measures such as skin conductance response. This methodological choice has been identified as a potential weakness by one reviewer, a critique that finds some justification within the traditional animal literature context. However, it is also pertinent to note that recent discussions within the field have begun to challenge the validity of fear conditioning as a reliable measure of fear. The reliance on subjective reports for assessing both loneliness and fear conditioning/extinction introduces complex issues that merit careful consideration. We strongly encourage the authors to address these critical points, alongside the detailed feedback provided by the reviewers, to enhance the manuscript's contribution to the field.

Reviewers' comments:

Reviewer's Responses to Questions

**Comments to the Author**

1. Is the manuscript technically sound, and do the data support the conclusions?

Reviewer #1: Yes

Reviewer #2: Yes

2. Has the statistical analysis been performed appropriately and rigorously? 

Reviewer #1: Yes

Reviewer #2: Yes

3. Have the authors made all data underlying the findings in their manuscript fully available?

Reviewer #1: No

Reviewer #2: Yes

4. Is the manuscript presented in an intelligible fashion and written in standard English?

Reviewer #1: Yes

Reviewer #2: Yes

5. Review Comments to the Author

Reviewer #1: This manuscript presents a large-scale study of the effects of loneliness on the extinction of evaluative fear learning. The authors extend with this work their previous findings with classical fear conditioning. The topic of the manuscript is of interest and importance to better understand how loneliness may have an impact on fear extinction processes and therefore also clinical fear exposure. The methods are sound and the manuscript is well written.

I just have a few minor points that I would like to see implemented:

- Abstract: Please add more information about the methods used. Given the large sample size used, it would also be good to see the final n used for the extinction analysis already here.

- Throughout the manuscript there are frequent references to manuscripts in preparation or submitted. I am not sure what Plos One's policy is on this, but I would prefer to see a reference to a pre-registration or preprint so that the reader can actually read the source of the reference. For example, in the Methods "(see: L. Lazar, EAH (author), & 181 NIE (author), in prep))

- Methods: "(...) protocols designed to examine the effects of a prosocial behavior intervention on feelings of well-being" There is no indication that the study was pre-registered. Was this analysis or the larger study pre-registered somewhere? If so, please provide a reference.

- It’s not clear what is used as the “9 target neutral images”. Are these also images from the IAPS with the same neutral content as the secondary neutral images used for the evaluative conditioning? Please clarify this in the methods.

- l.364, l.393 The justification of the results should be in the Discussion section only.

Reviewer #2: This paper studied the fear learning and extinction in high and low loneliness participants. And the authors found that the patterns of the extinction of the evaluative fear associations were different between the high and low individuals. This manuscript presents a compelling and well-executed study that significantly contributes to the field of social bonds on fear learning. The authors demonstrate a thorough understanding of the research topic and employ a rigorous methodology to investigate their research questions. Overall, the manuscript is well-written, engaging, and likely to generate interest among readers. However, there are still some questions in the manuscript. The details are as follows:

Major questions

1. In the Introduction, the authors mentioned the negative influence of perception of being socially isolated on mental health outcomes and they also mentioned that no study have invested the associations between loneliness and fear learning. The definition between perception of social isolation and social exclusion are very similar. According to previous study (Cacioppo&Cacioppo, 2016), Social exclusion has been defined broadly as “the experience of being kept apart from others physically or emotionally， Therefore, perception of social isolation, ostracism, and romantic rejection fall under the broad rubric of social exclusion”. Thus, Dou et al.(2020) have already investigated that the influences of social exclusion on fear learning in humans and found the participants after social exclusion inhibited fear conditioning compared with those after social inclusion. Surprisingly, I found the authors introduced lots of animal’s studies, rather than mentioned this closely related paper in humans in the introduction section.

2. In the introduction and discussion sections, the authors used a large amount sentence to interpret that the anxiety may play an important role in relation between loneliness and fear extinction. However, the author did not report any anxiety score for each participant. As the authors mentioned, the individuals with anxiety disorders showed a dysfunction patterns during the classical conditioning and extinction(Lissek et al., 2005; Duits et al., 2015). Whether the high loneliness groups showed higher anxiety scores compared to the low loneliness groups? Thus, whether the different patterns of extinction in high/low loneliness were also due to the differences of anxiety scores?

3. Similarly, the lots of previous studies(Cacioppo et al., 2006; Erzen & Çikrikci, 2018; Matthews et al., 2016; Singh & Misra, 2009) have already reported loneliness showed a close relationship with depression, even in different ages, such as youth, adult, and elderly. Moreover, previous research have reported individuals with depression showed dysfunction in affective processing, especially the reward learning(Admon & Pizzagalli, 2015). During this study, whether the depression showed a significant difference between the high and low loneliness group? And what role did the depression play between the loneliness and reward learning and extinction?

4. In the previous fear learning and extinction paradigm, compared to subjective ratings, some physiological responses to conditioned stimulus(CS) (such as skin conductance responses, SCR) were more widely used((Lonsdorf et al., 2017). Thus, the dependent variables in this manuscript were only the subjective ratings, which might influence accuracy of the results of the acquisition and extinction. I think this is a limitation in this manuscript.

5. Similarly, the habitation stage was also common adopted in the conditioning studies, which could reduce the extra variable evoked by the stimuli. However, the authors did not adopt the habitation stage in this study.

6. Although the influence of the different interventions (including prosocial behavior intervention, self-kindness internation, and control) are calculated, the influence of on the extinction might also mixed in the experiment, which might be better to add in the limitation section.

7. Although the authors set the cut-off for the loneliness or not, I am still curious about the correlation between UCLA scores and the ratings in both of the acquisition and extinction throughout the whole samples.

Minor questions

1. The figure of the procedure was not clear enough. What were the details in the acquisition section? Whether the CS and US presented on the screen at the same time or not?

2. In figures of acquisition and extinction, the error bar were missing, which could be added to improve the quality of the figures.

6. PLOS authors have the option to publish the peer review history of their article (what does this mean?). If published, this will include your full peer review and any attached files.

Reviewer #1: No

Reviewer #2: **Yes: **Haoran Dou

---

## [Author Response · Author response to Decision Letter 0]

21 Mar 2024

Dear Dr. Akaishi,

Thank you for giving us the opportunity to resubmit our manuscript along with changes based on your and the reviewer’s suggestions. We feel that the feedback we received has helped to make the findings reported more accessible and the manuscript stronger overall. Below, we have provided specific responses to the comments and concerns raised as well as the journal requirements.

Comments from the Editor: 

The manuscript has elicited positive feedback from both reviewers, who have acknowledged the commendable execution of the study and the articulation of the paper. Nonetheless, both reviewers have highlighted significant concerns regarding the manuscript's positioning within the extensive body of literature on loneliness and fear conditioning. A particular point of contention is the methodological approach adopted in the study, notably the preference for subjective reports over objective measures such as skin conductance response. This methodological choice has been identified as a potential weakness by one reviewer, a critique that finds some justification within the traditional animal literature context. However, it is also pertinent to note that recent discussions within the field have begun to challenge the validity of fear conditioning as a reliable measure of fear. The reliance on subjective reports for assessing both loneliness and fear conditioning/extinction introduces complex issues that merit careful consideration. We strongly encourage the authors to address these critical points, alongside the detailed feedback provided by the reviewers, to enhance the manuscript's contribution to the field.

We appreciate this feedback and would to note that we have addressed reviewer concerns regarding the use of subjective concerns during the evaluative conditioning procedures (please see the response to Reviewer 2, comment 4). With regard to recent discussions regarding the value and validity of fear conditioning, it is important to note that while different conceptualizations of the emotion of fear have been presented in the last decade—fear conditioning, and the outcomes that result from it, has a long and studied history that is not only reliable in both animals and humans but has been demonstrated to translate to both pathological processes as well as their treatments. These important and documented contributions of fear conditioning as a measure of fears and dysfunctional fears—as well as how to mediate these experiences—renders the continued study of fear conditioning an extremely important contribution to the field.

Comments to the Author:

1. Is the manuscript technically sound, and do the data support the conclusions?

Reviewer #1: Yes

Reviewer #2: Yes

2. Has the statistical analysis been performed appropriately and rigorously? 

Reviewer #1: Yes

Reviewer #2: Yes

3. Have the authors made all data underlying the findings in their manuscript fully available?

Reviewer #1: No

As noted, we were waiting until all papers stemming from this dataset were both in the publication process to make this data publicly available. However, regardless of their status, upon acceptance of this work we will make the data publicly available on the Harvard Dataverse.

Reviewer #2: Yes

4. Is the manuscript presented in an intelligible fashion and written in standard English?

Reviewer #1: Yes

Reviewer #2: Yes

Reviewer #1: 

This manuscript presents a large-scale study of the effects of loneliness on the extinction of evaluative fear learning. The authors extend with this work their previous findings with classical fear conditioning. The topic of the manuscript is of interest and importance to better understand how loneliness may have an impact on fear extinction processes and therefore also clinical fear exposure. The methods are sound and the manuscript is well written.

Thank you so much for this kind feedback. We very much appreciate your suggestions and have responded to each below.

I just have a few minor points that I would like to see implemented:

1. Abstract: Please add more information about the methods used. Given the large sample size used, it would also be good to see the final n used for the extinction analysis already here.

We have included more description of the evaluative conditioning methods used in the study as well as the final sample size used to assess the strength of fear conditioning in our analyses (please see lines 43-48).

2. Throughout the manuscript there are frequent references to manuscripts in preparation or submitted. I am not sure what Plos One's policy is on this, but I would prefer to see a reference to a pre-registration or preprint so that the reader can actually read the source of the reference. For example, in the Methods "(see: L. Lazar, EAH (author), & 181 NIE (author), in prep))

Unfortunately, these papers are in prep and were not pre-registered/do not yet have pre-prints to reference at this point. We are happy to remove these citations if the journal requests us to do so, however we do think they are important for the paper itself—especially the reference noted above which refers to the other paper that will be published based on the data collected during this study. We have updated this reference, as this other paper is closer to being submitted, and it is our hope that by the time of the publication of this manuscript, the other paper stemming from this data will be further along in the publication process and we can refer instead to a preprint.

3. Methods: "(...) protocols designed to examine the effects of a prosocial behavior intervention on feelings of well-being" There is no indication that the study was pre-registered. Was this analysis or the larger study pre-registered somewhere? If so, please provide a reference.

Unfortunately, although the hypotheses for this study were spelled out in the grant that funded it, neither this study nor the analyses used here were officially pre-registered so we cannot reference this in the current manuscript.

4. It’s not clear what is used as the “9 target neutral images”. Are these also images from the IAPS with the same neutral content as the secondary neutral images used for the evaluative conditioning? Please clarify this in the methods.

We have now added more detail regarding the 9 target neutral images (which were all images of everyday objects on a white background). Please see lines 238-240 in the revised manuscript.

5. l.364, l.393 The justification of the results should be in the Discussion section only.

We have removed the justification language from line 364 (now line 399), but were unable to determine which language the reviewer found inappropriate on line 393. We would be happy to make further adjustments with further clarification.

Reviewer #2: 

This paper studied the fear learning and extinction in high and low loneliness participants. And the authors found that the patterns of the extinction of the evaluative fear associations were different between the high and low individuals. This manuscript presents a compelling and well-executed study that significantly contributes to the field of social bonds on fear learning. The authors demonstrate a thorough understanding of the research topic and employ a rigorous methodology to investigate their research questions. Overall, the manuscript is well-written, engaging, and likely to generate interest among readers. 

We appreciate these encouraging words and that the reviewer feels this work represents a contribution to the field. We have addressed the reviewer’s suggestions to improve the paper below. 

However, there are still some questions in the manuscript. The details are as follows:

Major questions

1. In the Introduction, the authors mentioned the negative influence of perception of being socially isolated on mental health outcomes and they also mentioned that no study have invested the associations between loneliness and fear learning. The definition between perception of social isolation and social exclusion are very similar. According to previous study (Cacioppo&Cacioppo, 2016), Social exclusion has been defined broadly as “the experience of being kept apart from others physically or emotionally， Therefore, perception of social isolation, ostracism, and romantic rejection fall under the broad rubric of social exclusion”. Thus, Dou et al.(2020) have already investigated that the influences of social exclusion on fear learning in humans and found the participants after social exclusion inhibited fear conditioning compared with those after social inclusion. Surprisingly, I found the authors introduced lots of animal’s studies, rather than mentioned this closely related paper in humans in the introduction section.

The paper the reviewer points out is such an excellent paper and a very important contribution to the growing understanding of the role of social connection or disconnection in fear learning. In the case of subjective social isolation (loneliness) or objective social isolation, it is important to note that these experiences are defined by either a perceived lack of close relationships or an actual lack of relationships/contact that is chronic and globally experienced across an individual’s life—not by the acute and intentional removal of a relationship that is experienced in social exclusion. For this reason, we felt that these experiences were too dissimilar to compare across, even though they share similarities (just as it is difficult to compare across the effects of chronic and acute pain). However, it is clear that these studies all fall under the same umbrella of building knowledge of the powerful role of social connection on fear learning and therefore we have included discussion of this work in the introduction of our revised manuscript (please see lines 131-135). 

2. In the introduction and discussion sections, the authors used a large amount sentence to interpret that the anxiety may play an important role in relation between loneliness and fear extinction. However, the author did not report any anxiety score for each participant. As the authors mentioned, the individuals with anxiety disorders showed a dysfunction patterns during the classical conditioning and extinction (Lissek et al., 2005; Duits et al., 2015). Whether the high loneliness groups showed higher anxiety scores compared to the low loneliness groups? Thus, whether the different patterns of extinction in high/low loneliness were also due to the differences of anxiety scores?

The reviewers raise an important question here and in their next comment. We did collect anxiety and depression reports from participants (using the State/Trait Anxiety Scale and the Beck Depression Inventory, respectively) and examination of these scores revealed that, when split into low vs high groups (like those used to assess loneliness), neither anxiety nor depression showed any significant interaction with the change from acquisition to extinction in any condition (positive, neutral, or fearful), suggesting that neither of these experiences affect fear extinction in the way that loneliness was found to have done. Further, correlation analyses showed that neither anxiety nor depression had any significant relationship with change of ratings from acquisition to extinction, once again showing these scores don’t appear to influence fear extinction in the same way loneliness did in our results. 

3. Similarly, the lots of previous studies(Cacioppo et al., 2006; Erzen & Çikrikci, 2018; Matthews et al., 2016; Singh & Misra, 2009) have already reported loneliness showed a close relationship with depression, even in different ages, such as youth, adult, and elderly. Moreover, previous research have reported individuals with depression showed dysfunction in affective processing, especially the reward learning(Admon & Pizzagalli, 2015). During this study, whether the depression showed a significant difference between the high and low loneliness group? And what role did the depression play between the loneliness and reward learning and extinction?

As expected, we did find a significant difference in depression scores across the high and low lonely groups (p = .041), yet we found no effect of depression on fear extinction outcomes (please see response to above comment: Reviewer 2, comment 2). Interestingly, we did not find a significant difference in anxiety scores across the low and high lonely groups (p > .1). As with depression scores, we found no effect of anxiety on fear extinction outcomes (again, please see response to above comment: Reviewer 2, comment 2). 

4. In the previous fear learning and extinction paradigm, compared to subjective ratings, some physiological responses to conditioned stimulus(CS) (such as skin conductance responses, SCR) were more widely used((Lonsdorf et al., 2017). Thus, the dependent variables in this manuscript were only the subjective ratings, which might influence accuracy of the results of the acquisition and extinction. I think this is a limitation in this manuscript.

We appreciate the reviewer raising this issue. While subjective ratings and measurements of physiological responding are certainly different in nature, there is evidence regarding not only the value of subjective reporting but also the connection between this type of reporting/evaluative conditioning and the physiological measures often used in in-lab Pavlovian conditioning procedures. We have therefore added discussion of these important issues in the revised manuscript (see lines 170-179). However, we agree with the reviewer that evaluative conditioning on its own is not the same as in-person Pavlovian procedures that are often used to assess fear learning. Therefore, we have addressed this limitation and the need for future work to investigate the relationship between loneliness and non-evaluative, Pavlovian fear conditioning using non-reported, physiological measures in the discussion (please see lines 546-554 in the revised manuscript). 

5. Similarly, the habitation stage was also common adopted in the conditioning studies, which could reduce the extra variable evoked by the stimuli. However, the authors did not adopt the habitation stage in this study.

The reviewer raises an important question regarding the use of the habituation stage Indeed, the use of habituation for CSs is common in in-person conditioning procedures to ensure that participants are familiar with the conditional stimuli so that they no longer elicit physiological responding due to surprise at their appearing and/or their novelty. This also allows for examination of any pre-existing characteristics of any of the stimuli that continue to elicit differential responding even after familiarity is achieved. However, habituation is not always used and indeed can have drawbacks, including visual and attentional fatigue. Additionally, in cases when subjective reports, not physiology, are being measured, the need to reduce responding due to novelty is not necessary—for while novelty or lack of experience with images occurring on the screen might lead to incre

---

## [Decision Letter · Decision Letter 1]

3 May 2024

Loneliness and the persistence of fear: Perceived social isolation reduces evaluative fear extinction

PONE-D-24-02974R1

Dear Dr. Eisenberger,

We’re pleased to inform you that your manuscript has been judged scientifically suitable for publication and will be formally accepted for publication once it meets all outstanding technical requirements.

Kind regards,

Rei Akaishi

Academic Editor

PLOS ONE

Additional Editor Comments (optional):

The revisions submitted by the authors have been both thorough and meticulous. The manuscript now satisfactorily addresses all the concerns raised during the initial review. The authors have diligently incorporated the suggested changes, enhancing both clarity and coherence throughout the document. Moreover, the revisions reflect a deep and comprehensive understanding of the subject matter, as demonstrated by the nuanced responses to all inquiries. Overall, the careful and detailed nature of these revisions has significantly improved the manuscript, making it ready for publication.

Reviewers' comments:

Reviewer's Responses to Questions

**Comments to the Author**

1. If the authors have adequately addressed your comments raised in a previous round of review and you feel that this manuscript is now acceptable for publication, you may indicate that here to bypass the “Comments to the Author” section, enter your conflict of interest statement in the “Confidential to Editor” section, and submit your "Accept" recommendation.

Reviewer #1: All comments have been addressed

Reviewer #2: All comments have been addressed

2. Is the manuscript technically sound, and do the data support the conclusions?

Reviewer #1: (No Response)

Reviewer #2: Yes

3. Has the statistical analysis been performed appropriately and rigorously? 

Reviewer #1: (No Response)

Reviewer #2: Yes

4. Have the authors made all data underlying the findings in their manuscript fully available?

Reviewer #1: (No Response)

Reviewer #2: Yes

5. Is the manuscript presented in an intelligible fashion and written in standard English?

Reviewer #1: (No Response)

Reviewer #2: Yes

6. Review Comments to the Author

Reviewer #1: (No Response)

Reviewer #2: The revisions made by the authors have been thorough and detailed. The manuscript now adequately addresses all the queries raised during the initial review process. The author has diligently incorporated the suggested changes, providing clarity and coherence throughout the document. Additionally, the revisions demonstrate a comprehensive understanding of the subject matter, as evidenced by the nuanced responses to all inquiries. Overall, the meticulous nature of the revisions ensures that the manuscript is now significantly improved and ready for further consideration.

7. PLOS authors have the option to publish the peer review history of their article (what does this mean?). If published, this will include your full peer review and any attached files.

Reviewer #1: No

Reviewer #2: **Yes: **Haoran Dou

---

## [Editor Report · Acceptance letter]

9 Aug 2024

PONE-D-24-02974R1 

PLOS ONE

Dear Dr. Eisenberger, 

I'm pleased to inform you that your manuscript has been deemed suitable for publication in PLOS ONE. Congratulations! Your manuscript is now being handed over to our production team.

Kind regards, 

on behalf of

Dr. Rei Akaishi 

Academic Editor

PLOS ONE